# Interaction of *Thalassia testudinum* Metabolites with Cytochrome P450 Enzymes and Its Effects on Benzo(a)pyrene-Induced Mutagenicity

**DOI:** 10.3390/md18110566

**Published:** 2020-11-19

**Authors:** Livan Delgado-Roche, Rebeca Santes-Palacios, José A. Herrera, Sandra L. Hernández, Mario Riera, Miguel D. Fernández, Fernando Mesta, Gabino Garrido, Idania Rodeiro, Jesús Javier Espinosa-Aguirre

**Affiliations:** 1Departamento de Farmacología, Instituto de Ciencias del Mar (ICIMAR), Loma 14, Alturas del Vedado, Plaza de la Revolución, La Habana 10600, Cuba; ldelgado@liomont.com.mx (L.D.-R.); mario@icimar.cu (M.R.); migueldavid@cebimar.cu (M.D.F.); 2Dirección Médica, Laboratorios Liomont S.A. de C.V., Carretera México-Toluca 5420, Ciudad de México 05320, Mexico; 3Laboratorio de Toxicología Genética, Instituto Nacional de Pediatría, Insurgentes Sur 3700, Insurgentes Cuicuilco, Ciudad de México 04530, Mexico; rebeca.santes@inp.mx; 4Instituto de Ciencia y Tecnología de Materiales (IMRE), Universidad de La Habana, Zapata y G, Vedado, Plaza de la Revolución, La Habana 10400, Cuba; jose@imre.oc.uh.cu; 5Departamento de Medicina Genómica y Toxicología Ambiental, Instituto de Investigaciones Biomédicas, Universidad Nacional Autónoma de México (UNAM), Ciudad Universitaria, Ciudad de México 04510, Mexico; slhernandez@biomedicas.unam.mx; 6Escuela Nacional de Medicina y Homeopatía, Instituto Politécnico Nacional, Guillermo Massieu Helguera 239, Ciudad de México 07320, Mexico; lmestac1800@alumno.ipn.mx; 7Departamento de Ciencias Farmacéuticas, Facultad de Ciencias, Universidad Católica del Norte, Angamos 0610, Antofagasta 1240000, Chile; gabino.garrido@ucn.cl

**Keywords:** *Thalassia testudinum*, thalassiolin B, polyphenols, CYP1A1, benzo[a]pyrene, chemoprevention

## Abstract

The aim of the present work was to evaluate the effects of *Thalassia testudinum* hydroethanolic extract, its polyphenolic fraction and thalassiolin B on the activity of phase I metabolizing enzymes as well as their antimutagenic effects. Spectrofluorometric techniques were used to evaluate the effect of tested products on rat and human CYP1A and CYP2B activity. The antimutagenic effect of tested products was evaluated in benzo[a]pyrene (BP)-induced mutagenicity assay by an Ames test. Finally, the antimutagenic effect of *Thalassia testudinum* (100 mg/kg) was assessed in BP-induced mutagenesis in mice. The tested products significantly (*p* < 0.05) inhibit rat CYP1A1 activity, acting as mixed-type inhibitors of rat CYP1A1 (Ki = 54.16 ± 9.09 μg/mL, 5.96 ± 1.55 μg/mL and 3.05 ± 0.89 μg/mL, respectively). Inhibition of human CYP1A1 was also observed (Ki = 197.1 ± 63.40 μg/mL and 203.10 ± 17.29 μg/mL for the polyphenolic fraction and for thalassiolin B, respectively). In addition, the evaluated products significantly inhibit (*p* < 0.05) BP-induced mutagenicity in vitro. Furthermore, oral doses of *Thalassia testudinum* (100 mg/kg) significantly reduced (*p* < 0.05) the BP-induced micronuclei and oxidative damage, together with an increase of reduced glutathione, in mice. In summary, *Thalassia testudinum* metabolites exhibit antigenotoxic activity mediated, at least, by the inhibition of CYP1A1-mediated BP biotransformation, arresting the oxidative and mutagenic damage. Thus, the metabolites of *T. testudinum* may represent a potential source of chemopreventive compounds for the adjuvant therapy of cancer.

## 1. Introduction

Air pollution-related diseases represent a major environmental problem affecting health worldwide. Outdoor and indoor pollution has been tightly associated with the incidence of respiratory diseases, including lung cancer. In accordance with WHO estimates, lung cancer causes about 6% of premature deaths related to outdoor air pollution, as well as tobacco consumption [1]. Polycyclic aromatic hydrocarbons (PAHs) are pollutants widely distributed in the environment as a result of organic matter incomplete combustion. Furthermore, PAHs are present in commercial products consumed by humans (e.g., tobacco) [2]. Benzo[a]pyrene (BP), one of the main PAH air pollutants, is metabolized by CYP1A1 and CYP1B1 enzymes and it is biotransformed into carcinogenic (±)-B[a]P-r-7, t-8-dihydrodiol-t-9,10 (BPDE) epoxide species [2,3]. BPDE reacts with DNA to produce BPDE-N2-deoxyguanosine adducts, promoting DNA mutations and carcinogenesis [4,5,6,7]. The presence of BPDE-DNA adducts in human lung cells has been well documented to be related to the initiation of pulmonary cancer [5,8].

Advances in the pharmaceutical industry improve cancer treatment; however, the discovery of new effective chemopreventive agents is still necessary. Cancer chemoprevention by natural or synthetic agents capable of avoiding, reversing or suppressing carcinogenic progression has become a plausible strategy to arrest cancer mortality [9]. There are several classes of cancer chemopreventive agents including blocking agents, which act at the initiation stage of carcinogenesis by inhibiting pro-carcinogen activating enzymes by inducing carcinogen-detoxifying enzymes, by enhancing antioxidant activity or by inducing DNA repair enzymes [10]. Cytochrome P450 (CYP450) enzymes are a superfamily of hemoproteins that catalyze the biotransformation of not only a wide array of drugs and endogenous substances, but also the bioactivation of many pro-carcinogens [11]. Consequently, specific CYP enzymes have been identified as potential targets for cancer chemoprevention [12].

Polyphenols exert a wide range of beneficial effects beyond their antioxidant and anti-inflammatory properties [13,14,15]. Data evidence from in vitro [16,17,18,19,20] and in vivo studies [21,22,23,24] suggest the chemoprotective role of polyphenols against lung carcinogenesis probably resulting from three main mechanisms: antioxidant activity, regulation of phase I and II enzymes and regulation of cell survival pathways [24].

Marine plants are a potential source of secondary metabolites with beneficial properties [25,26,27,28,29,30,31,32]. *Thalassia testudinum* seagrass grows abundantly in the Caribbean Sea, particularly in the Cuban coasts. A previous study reports sulfated glycoside flavone thalassiolin B (TB) (chrysoeriol7-β-d-glucopyranosyl-2”-sulphate, Figure 1) as the most abundant bioactive component within the *T. testudinum* crude hydroethanolic extract (Th) [33]. Other phenolic compounds have been identified in the extract, including apigenin-7-O-β-d-glucopyranosyl-2′′-sulfate (thalassiolin C), chrysoeriol-7-O-β-d-glucopyranoside, apigenin-7-O-β-d-glucopyranoside, dihydroxy-3′,4′-dimethoxyflavone 7-O-β-d-glucopyranoside, luteolin-3′-sulphate, chrysoeriol and apigenin [34]. Th shows in vitro scavenger activity for ^•^OH, RO_2_^•^, O_2_^•−^ and DPPH^•^ free radicals and in vivo antioxidant effects against brain and liver induced-lipid peroxidation in mice [34,35]. In addition, Th shows acute anti-inflammatory effects in mice [36] and it displays selective anti-proliferative activity against cancer cells compared to normal cells [37]. Besides, the extract also inhibits drug efflux by ABCG2/breast cancer resistance protein (BCRP) and ABCB1/P-glycoprotein (MDR1 gene), increasing intracellular accumulation of anticancer agents [38,39]. Thus, the marine angiosperm *T. testudinum* has been considered a natural source of potential antitumor agents.

On the other hand, Th modulates the activity of different isoforms of P450 system, including CYP1A and 2B families [38,40]; however, these interactions are not well characterized yet. As CYP1A and 2B subfamilies are involved in the metabolism of several mutagens and carcinogens, the enzymatic inhibition could be associated with decreased carcinogenic risk. Thus, the aim of the present work was to further characterize the effects of *T. testudinum* extract and its polyphenolic components (polyphenolic fraction of the hydroethanolic extract, PF) on CYP1A and CYP2B enzymatic activity, an also to evaluate the effects of Th on BP-induced mutagenicity.

## 2. Results

### 2.1. Tested Compounds Modulate Rat CYP1A But Not CYPB2 Activity

The enzymatic activity of CYP1A1/2 CYP2B1/2 was measured in rat liver microsomes in the presence, or not, of Th, PF or TB. Test products shown no interference with the fluorescence of resorufin even at the highest concentration tested. No appreciable changes in the activity of both CYP2B isoforms were observed (data not shown). In contrast, Th, PF and TB modulated the rat CYP1A activities as shown in Table 1. The enzymatic activity of both CYP1A1 and CYP1A2 was modulated by the test natural products; however, CYP1A1 was more sensitive than CYP1A2. The PF and TB showed a significant (*p* < 0.05) higher inhibitory effect than the crude extract (Th) on CYP1A isoforms; meanwhile, the Th showed no significant inhibition for CYP1A2.

### 2.2. T. testudinum Extract, Polyphenolic Fraction and Thalassiolin B Are CYP1A1 Mixed-Type Inhibitors

Once CYP1A1 was identified as the most sensitive enzyme, kinetics experiments were performed in order to elucidate the type of inhibition induced by PF and TB. Demethylation of EROD in the presence of rat liver microsomes showed typical Michaelis–Menten kinetics for evaluated products (Figure 2A–C). Using non-linear regression and a Lineweaver–Burk plot, it was determined that Th, PF and TB are mixed-type inhibitors for rat CYP1A1 (Figure 2D–F) with constants of inhibition (Ki) of 54.16 ± 9.09 μg/mL, 5.96 ± 1.55 μg/mL and 3.05 ± 0.89 μg/mL, respectively (Table 2). In a mixed-type inhibition model, affinity changes of the enzyme for the substrate in presence of an inhibitor is determined by the parameter alpha (α), which was 8.66 ± 2.82 for Th, 370.60 ± 56.86 for PF and 3.65 ± 0.86 for TB (Table 2).

### 2.3. Polyphenolic Fraction and Thalassiolin B Modulate the Human CYP1A1 Activity

Taking into account that rat CYP1A1 was more sensitive than the CYP1A2 isoform, the effect of tested products on human recombinant CYP1A1 activity was evaluated. The results showed a significant (*p* < 0.05) inhibition of human recombinant CYP1A1 by PF and TB, while Th only exhibited a slight inhibitory effect on enzyme activity at the highest concentration (Figure 3A). The biochemical characterization of human CYP1A1 inhibition resulted in a mixed-type for PF and non-competitive inhibition for TB (Figure 3B–E). A moderate inhibitory potential was found for PF and TB on human CYP1A1 with Ki of 197.10 ± 63.40 μg/mL and 203.10 ± 17.29 μg/mL, respectively (Table 2). These results showed a potential inhibitory effect of phase I carcinogen-metabolizing enzymes CYP1A1 by *T. testudinum* metabolites.

### 2.4. Antimutagenic Effect of T. testudinum Extract, Polyphenolic Fraction and Thalassiolin B against Benzo[a]pyrene-Induced Mutagenicity in S. typhimurium

The bioactivation of BP by CYP1A1 leads to carcinogenic effects [2,5]. Since Th, PF and TB exert inhibitory effects on both rat and human CYP1A1 activity, we explored the potential anti-mutagenic effect of these compounds by using the Ames test. First, we evaluate the cytotoxicity of test products (up to 1000 µg/mL) on *S. typhimurium*. The frequency of spontaneous reversion in controls (S9 plus vehicle) did not differ from the historically controls of our laboratory for *S. typhimurium*. Therefore, the test products are not cytotoxic for *S. typhimurium*, whereas BP induced a significantly (*p* < 0.05) increase of revertant frequency. Interestingly, the number of revertant colonies decreased in a dose-dependent fashion in presence of test natural products. The inhibition percentage of BP-induced mutagenicity in presence of S_9_ activation mixture achieved 27% for the highest tested dose of the extract (1000 µg/mL), 34% for PF (500 µg/mL) and 33% for TB (400 µg/mL) (Table 3). These results suggested that Th, PF and TB possess antimutagenic effects under these experimental conditions.

### 2.5. T. testudinum Extract Reduces Oxidative Damage and Micronuclei Formation in BP-Exposed Mice

The capacity of Th to reduce BP-induced oxidative damage in mice was evaluated by measuring the serum levels of malondialdehyde (MDA), advanced oxidation protein products (AOPP) and reduced glutathione (GSH). As we expect, in Th pre-treated mice, the levels of MDA and AOPP were significantly lower (*p* < 0.05) than in BP control animals. In accordance with the reduction of oxidative damage, GSH level was significantly increased (*p* < 0.05) in Th pre-treated animals (Table 4). Additionally, BP induced a significant micronuclei formation in bone marrow of BP-exposed animals compared to cells from control animals (vehicle). Meanwhile, in Th pre-treated mice it was found a significant reduction (*p* < 0.05) of micronuclei formation (Table 5). These results indicate a protective effect by the extract obtained from *T. testudinum* leaves against BP-induced DNA damage together with a modulation of systemic oxidative stress in mice.

## 3. Discussion

There is evidence on the inhibitory effects of plant-derived phenols in PAH-induced mutagenesis and carcinogenesis [41,42]. The antimutagenic and chemopreventive properties of polyphenols have been associated with modulation of CYP450-mediated metabolism of mutagens, as well as to the interaction with active mutagenic metabolites [43]. It has been suggested that flavones which contain free 5- and 7-hydroxyls are potent inhibitors of cytochrome CYP1Al/2 [44]. Therefore, they may be useful as chemopreventive agents against PAH-induced carcinogenesis. The extract obtained from the leaves of *T. testudinum* marine plant is rich in flavonoids and other polyphenols (29.5% ± 1.2% total polyphenols, proanthocyanidins 21.0% ± 2.3%, total flavonoids 4.6% ± 0.2%, expressed as g per 100 g of the dry extract, % *w/w*) suggesting a potential inhibitory capacity on CYPs enzymatic activity. Derivate of *T. testudinum* are under preclinical investigation as new nutraceutical with promising active pharmacological effects [45,46,47]. The *T. testudinum* extract constitutes a potential source of chemopreventive agents because of its effectiveness as anti-inflammatory and antioxidant, which has been demonstrated previously [33,34,35,45,46,47].

In an earlier report, we describe the inhibitory effects of the Th and TB on the CYP1A1 activity in human hepatocytes [38]. The current work was aimed to further understanding of the effects of Th, PF and TB on CYP1A activity. This work let us evaluate the role of polyphenolic constituents of the extract, as substances responsible of its bioactive properties, in particular on phase I metabolizing enzymes.

Reduced O-Dealkylation of 7-ethoxyresorufin and 7-methoxyresorufin demonstrated the capacity of tested products to inhibit rat and human CYP1A1/2 activities. The polyphenolic-rich fraction isolated from the extract and TB revealed to be more active than Th, which in part supported our early hypothesis. All the natural products showed a mixed-type inhibition on rat CYP1A1, describing typical Michaelis–Menten kinetics, meanwhile exhibiting a differential response on human CYP1A1. Th did not inhibit the human CYP1A1 isoform. However, the PF and TB showed relevant inhibitory effects with differences in their kinetics and inhibition mechanism. PF acted as a mixed-type inhibitor, while TB behaved as non-competitive inhibition kinetics. These results were in agreement with previous reports on interspecies differences regarding the inhibitory potency of natural compounds [47,48] and also provided additional evidence on the interactions of polyphenols with CYP450 system.

Other naturally occurring polyphenolic compounds, such as galangin (3,5,7-trihydroxyflavone) and apigenin (5,7,4-trihydroxyflavone) also inhibit CYP1A1 and CYP1A2 activities [49]. Reported values of apparent Ki for the inhibition of CYP1A1by galangin is 0.015 µM [50] and 0.32 µM apigenin [51]. On the other hand, there are fewer reports regarding the inhibition of CYP1B1 catalytic activity by naturally occurring compounds than CYP1A1 and CYP1A2. In fact, the extract and its polyphenolic components did not modulate CYP2B family activity under our experimental conditions.

The capacity of polyphenols and flavonoids to inhibit CYP1A1/2 isoforms has been proposed as the main mechanism supporting the antimutagenic and chemopreventive activities of these natural compounds [52]. Metabolic activation from BP to BPDE is believed to be essential for the mutagenic and carcinogenic properties of this pollutant [53,54]. CYP1Al plays an important role in the biotransformation/activation of BP [55,56]. Thus, we assessed the antimutagenic properties of the extract and its derivate in *S. typhimurium* TA98 by the mutagenesis assay of Maron and Ames (1983) [57]. The aim of this assay was to explore the biological consequences of the CYP1A modulation by the tested products. BP bioactivation was achieved by phenobarbital-induced rat liver S_9_ fraction. It is widely accepted that the Ames assay is useful for correlating in vitro mutagenesis and in vivo carcinogenicity in animals and humans. Th, PF and TB exerted a protective effect against BP-induced mutagenicity in *S. typhimurium*, evidenced here by the reduction of His^+^ revertant colonies per plate, which suggested a potential antimutagenic activity of this marine organism.

Interestingly, in a previous report we observed a significant increase in BP mutagenicity after incubation with S_9_ fractions obtained from rats, orally treated during 10 days with doses of 200 and 400 mg/kg of Th; however, mutant colonies were reduced at low evaluated doses (20 mg/kg) [40]. We also reported a differential response of CYP1A1, 1.5-fold increase after treatment with 200 mg/kg of Th, a discrete increase at 400 mg/kg dose and no significant changes at the low dose of *T. testudinum* extract. Differences in CYP activity suggest that polyphenol concentration could be a critical factor mediating the interaction between *T. testudinum* extract and CYP1A1 activity or BP mutagenicity. The present work may support this hypothesis, since BP-induced DNA damage, micronucleus frequency and oxidative damage diminished in orally pre-treated mice with 100 mg/kg of Th. These results are in agreement with the in vitro Ames test results and contrast with previous reports of our group in which a high dose of Th was assessed.

Antioxidant and anti-inflammatory properties of *T. testudinum* extract have been also observed under similar experimental conditions [46,58]. Accordingly, *T. testudinum* extract could protect DNA by different mechanisms, where antioxidant effects and the modulation of BP bioactivation should be conjugated. The capacity of *T. testudinum* components to inhibit CYP1A enzymatic activity could certainly be involved in the observed chemoprotective effect, but the alternative hypothesis which considers that Th components may act as scavengers that bind reactive metabolites of BP must not be discarded (assessed in ongoing studies). Previously, we demonstrated that the extract protects rat hepatocytes from terbutil-hidroperoxide-induced GSH depletion, together with higher catalase and superoxide dismutase activities compared with controls [35]. Th also showed protective effects against ethanol-, carbon tetrachloride- and lipopolysaccharide-induced cytotoxicity in rat hepatocytes [35]. Therefore, it seems that a combination of different mechanisms might be involved in the complex interaction of *T. testudinum* extract and the ADME cellular process, which in turn has influence its bioactive properties.

DNA oxidative damage is critical for BP in vivo toxicity [59,60,61]. It has been also reported that BP induces genetic lesions such as DNA single-strand breaks, DNA–protein cross-links and chromosomal aberrations [60]. The protective effects of Th against acute exposure to BP were corroborated here by mean the micronucleus assay. As we expected, Th reduced not only the BP-oxidative damage but also the micronucleus formation, meanwhile an increase of antioxidant defenses, such as GSH levels, was observed.

These results altogether support new findings on the modulation of CYP system and particularly CYP1A1 by *T. testudinum* polyphenols. As mentioned before, CYP1A1/2 activity plays a significant role in the activation/detoxification balance of pro-carcinogens in hepatic and extrahepatic tissues. Therefore, the inhibitory effects of polyphenols present in *T. testudinum* marine plant on CYP1A activity may explain, in part, the antimutagenic and/or chemopreventive properties of these natural products.

## 4. Materials and Methods

### 4.1. Materials and Reagents

Beta-naphtoflavone (β-NF), 7-ethoxyresorufin (EROD), methoxyresorufin (MROD), benzyloxiresorufin (BROD), penthoxyresorufin (PROD), resorufin and benzo[a]pyrene (BP) were purchased from Sigma (St. Louis MO, USA). Culture media were obtained from BD Difco (Trenton, NJ, USA). The *Escherichia coli* DH5α recombinant and *Salmonella typhimurium* strains were kindly provided by Dr. Peter Guengerich (Vanderbilt University, Nashville, TN, USA).

### 4.2. Vegetal Material

*Thalassia testudinum* (Banks and Soland ex. Koenig) was collected during March 2017 in “Guanabo” beach (22°05′45″ N, 82°27′15″ W). The specimen was identified by Dr. J.A. Areces (Institute of Oceanology, Havana, Cuba) and a voucher sample (No. IdO40) is deposited in the herbarium of the Cuban National Aquarium. The leaves were washed with water to eliminate sediments and the excess of salt; then, the plant material was dried at room temperature. Whole dry and ground *T. testudinum* leaves (840 g) were continuously extracted with ethanol-H_2_O (50:50, *v/v*) during 24 h at room temperature. The extract was filtered and concentrated under reduced pressure and temperature (40 °C) to yield 54 g of crude extract (Th).

Th was used as a starting material to obtain a polyphenol-enriched fraction (PF), where non-polar components were excluded by chloroform extraction (1:10, w/v). The resultant PF was filtered and dried at room temperature. Later, TB (1-chrysoeriol 7-β-d-glucopyranosyl-2″-sulphate) was isolated by electrospray ionization mass spectrometry (ESIMS) with an [M − H]− ion at m/z = 541. The structure was confirmed by spectroscopic analysis (^1^H and ^13^C nuclear magnetic resonance) as shown in Figure 1, and compared to reported data [38].

### 4.3. Rat Liver S9 and Microsomal Fraction Obtaining

The S9 fraction was obtained as previously described [57]. To obtain the microsomal fraction, S_9_ was split into 1 mL aliquots and centrifuged at 100,000× *g* and 4 °C per 60 min. The pellet was resuspended in 0.1 M phosphate buffer (pH 7.4) plus 0.25 M sucrose and it was centrifuged again at 100,000× *g* at 4 °C per 60 min. The pellet (microsomal fraction) was resuspended in 0.1 M phosphate buffer (pH 7.4), 1 mM EDTA, 0.1 mM dithiothreitol and 20% *v/v* glycerol and stored at −80 °C until use.

### 4.4. Bacterial Membrane Fraction Obtaining

The isolation of membrane fractions from *Escherichia coli* expressing a recombinant human CYP1A1 was performed as previously described [62,63]. Briefly, overnight cultured *E. coli* DH5α were diluted 1:100 in Terrific Broth/ampicillin (100 μg/mL) medium containing 1 mM isopropyl b-d thiogalactoside, 0.5 mM aminolevulinic acid, 1 mM thiamine and trace salts. Bacteria cultures were grown during 24 h at 30 °C and 150 rpm shaking. Thereafter, *E. coli* membrane fractions were isolated from the whole pellets by serial d ultracentrifugation steps.

### 4.5. Enzymatic Activity Assays

#### 4.5.1. CYP1A1 and CYP1A2 Activities

CYP enzymatic activities were evaluated in rat liver microsomes and in bacterial membrane fraction by spectrofluorometric assay as previously described [48,64], with slight modifications. The reaction mixtures contained: rat liver microsomes (80 µg) or bacterial membrane fraction (40 µg), EROD (1 µM) or MROD (5 µM) substrate and 2.5 to 50 µg/mL of test products (Th, PF or TB). Then, buffer solution (Tris-HCl (50 mM) and MgCl2 (25 mM), pH = 7.6) was added to reaction mixtures and incubated 3 min at 37 °C. The reaction started by the addition of NADPH (0.5 mM). The fluorescence units were registered at 20 s intervals during 15 min in a hybrid multi-mode microplate reader (Synergy H4, Biotech). Finally, CYPs activities were calculated from a resorufin standard curve (5–50 pmol/mL). Interference of tested products with the fluorescence of resorufin was also carried out.

#### 4.5.2. CYP2B1 and CYP2B2 Activities

The activities of CYP2B1-related penthoxyresorufin O-dealkylase (PROD) and CYP2B2-related benzyloxyresorufin O-dealkylase (BROD) were determined in rat liver microsomes by spectrofluorometric techniques as previously described [64], with minor modifications. The assays were conducted under similar conditions previously described for CYP1A activity.

### 4.6. Kinetic Analysis of Enzyme Inhibition

Different concentrations of substrates (0.32–10 μM of EROD) mixed with rat liver microsomes (80 µg) or *E. coli* membrane fraction (40 µg) were used for enzymatic kinetics assays in presence or absence of the tested products. Kinetic constants were obtained by a nonlinear regression analysis of experimental data fitted to the Michaelis–Menten equation with competitive, non-competitive and mixed-type inhibition models (GraphPad Prism version 6 software). Kinetic analysis was also shown by using the Lineweaver–Burk plot.

### 4.7. Ames Test

The effects of the tested products against BP-induced mutagenicity was assessed according to Maron and Ames [57], in *Salmonella typhimurium* strain TA98 and rat liver S_9_ fraction. The reaction mixture contained 0.6% agar, 0.5% NaCl, 0.5 mM biotin and 0.05 mM L-histidine. A solution of the tested products (100 µL) at different concentrations (Th: 10, 100, 1000 µg/mL; PF: 5, 50, 500 µg/mL; TB: 400 µg/mL), overnight cultured *S. typhimurium* (10^8^ cells), S9 (500 µL) and benzo(a)pyrene (10 µg/plate) were added to the reaction mixture (2 mL). Phosphate buffer (500 µL) was used instead of the S_9_ fraction as negative control. Afterward, the plates were incubated at 37 °C for 48 h and the number of revertant colonies (His+) was quantified. The inhibition percentage of BP-induced mutagenicity was calculated as (1-(number colonies/plates with mutagen plus the tested products)/(number colonies/plate with just mutagen)) × 100%.

### 4.8. Effects of the T. testudinum Extract against BP-Induced DNA Damage in Mice

Balb/c male mice (20–25 g) were obtained from Centro para la Producción de Animales de Laboratorio (CENPALAB, Havana, Cuba). The animals were adapted to standard conditions (temperature: 20 ± 2 °C, humidity: 40–60%, 12 h light/dark cycle) during a week. Mice were fed with a standard diet and water ad libitum. Experimental procedures were carried out in accordance with European regulations on animal protection (Directive 86/609), and the Guide for the Care and Use of Laboratory Animals as adopted and promulgated by the US National Institute of Health (NIH Publication № 85–23, revised 1996). The experimental protocol was approved by the Institutional Animal Care and Ethical Committee from the Institute of Marine Sciences (ICIMAR), Havana, Cuba (Protocol number 1805, Date of approval: 28 March 2018). Four groups were included in the study (five animals per group). The first group was animals received seven daily oral doses of distilled water and one dose of oil (BP vehicle). The second group only received one oral dose of benzo(a)pyrene (250 mg/kg), the last day of the experiment. The third group was orally treated with Th aqueous solution (100 mg/kg) for seven days and one hour after the last administration of the extract, animals received an oral dose of BP (250 mg/kg). After 24 h, all the animals were sacrificed, and blood samples were obtained by cardiac puncture and centrifuged at 3000× *g* for 10 min, at 4 °C. Serum was collected and stored at −80 °C until use.

#### 4.8.1. Oxidative Stress Biomarkers Determination

Serum markers of oxidative stress were determined in mice. Malondialdehyde (MDA) content was determined as previously described [65]. Reduced glutathione (GSH) levels were measured at 412 nm after precipitation of thiol proteins by using the Ellman‘s reagent (5,5‘dithiobis-2-nitrobenzoic acid, Sigma, Burbank, CA, USA) in accordance with Sedlak and Lindsay [66]. The advanced oxidation protein products (AOPP) were quantified with potassium iodide (1.16 M) followed by the addition of acetic acid. The absorbance was immediately read at 340 nm. AOPP concentration was expressed as μM of chloramine-T. Concentration of oxidative stress biomarkers was expressed per mg of protein.

#### 4.8.2. Micronuclei Formation Determination

After mice euthanasia, the femurs were removed, and its proximal end was shortened until the marrow canal became visible. One milliliter of serum was introduced into the bone canal and the marrow was aspirated and flushed several times. Cells were centrifuged at 1000 rpm for 5 min at 4 °C. Afterward, they were fixed in methanol and stained with Giemsa 5% (*v/v*) for 12 min. The presence of micronuclei was determined in a sample of 2000 polychromatic erythrocytes (PCE). Normochromatic erythrocytes (NCE) were also scored in 200 erythrocytes samples to determine the PCE/NCE ratio [67].

### 4.9. Statistical Analysis

Statistical analyses were performed with GraphPad Prism 5.0 (GraphPad, La Jolla, CA, USA). Revertants/plate, enzymatic activities, MDA, GST, AOPP levels, and micronucleus data were expressed as mean ± SD values. For multiple mean comparisons was used a one-way ANOVA followed by Dunnett or Tukey non-parametric tests. The level of statistical significance was set to * *p* < 0.05, ** *p* < 0.01, or *** *p* < 0.001.

## 5. Conclusions

In summary, our results suggest that in vitro and in vivo antimutagenicity effects of *Thalassia testudinum* extract and its polyphenolic components may be mediated, at least, by the inhibitory effect on phase I metabolizing enzymes, in particular the CYP1A family. As consequence, reduced oxidative stress and mutagenic effects were observed. The results altogether contribute to support the potential of *T. testudinum* as a source of chemoprotective compounds against air pollution-mediated carcinogenesis.

## Figures and Tables

**Figure 1 marinedrugs-18-00566-f001:**
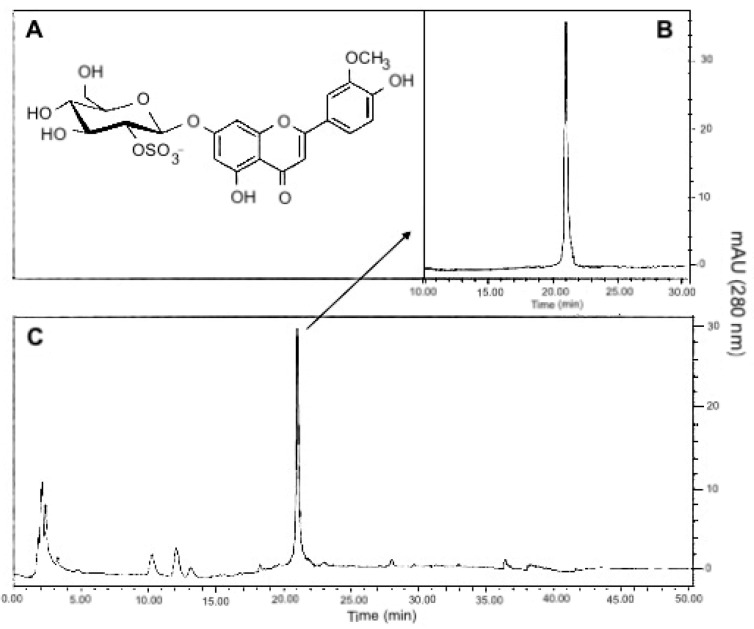
Chromatographic profile of thalassiolin B isolated from *T. testudinum* hydroethanolic extract. (**A**) Chemical structure of thalassiolin B (chrysoeriol 7-β-d-glucopyranosyl-2”-sulphate), the main component of *T. testudinum* extract. (**B**) HPLC of thalassiolin B standard. (**C**) HPLC profile of *T. testudinum* hydroethanolic extract. The authors have the right to use this figure.

**Figure 2 marinedrugs-18-00566-f002:**
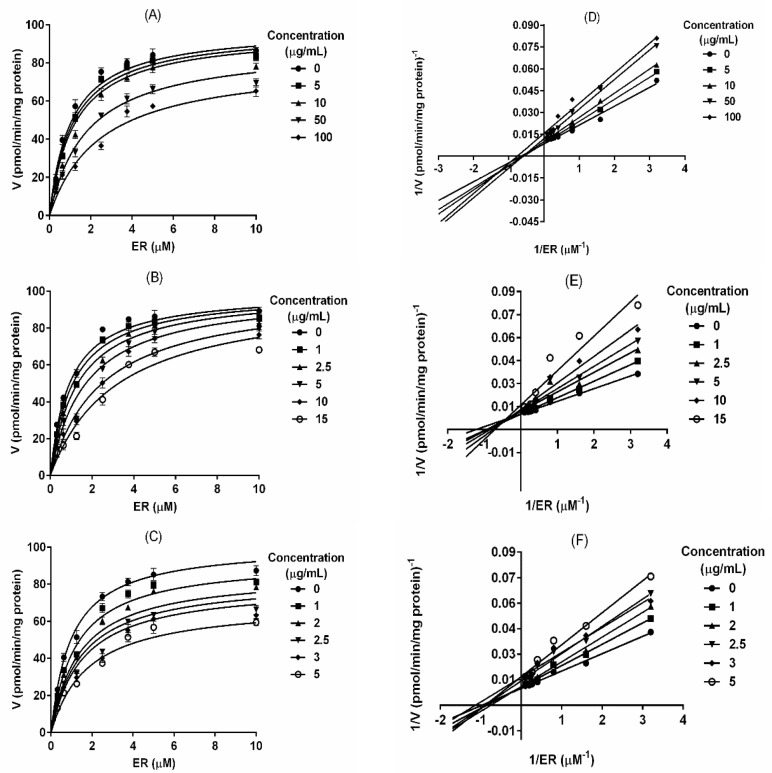
Inhibition kinetics of rat CYP1A1 by *T. testudinum* extract, polyphenolic fraction and Thalassiolin B. The fluorescence was recorded every 15 s during 15 min; reactions consisted in 80 µg protein, 0.32–10 μM 7-ethoxyresorufin, and 50 mM NADPH. For inhibition assays, the test products were added at different concentrations to the reaction mixture. Each point in (**A**–**C**) represents the mean ± SD from three independent experiments. (**D**–**F**) Lineweaver-Burk plot analyses were done to obtain the kinetic parameters. (**A**,**D**) *T. testudinum* extract (Th); (**B**,**E**) polyphenolic fraction (PF); (**C**,**F**) thalassiolin B (TB).

**Figure 3 marinedrugs-18-00566-f003:**
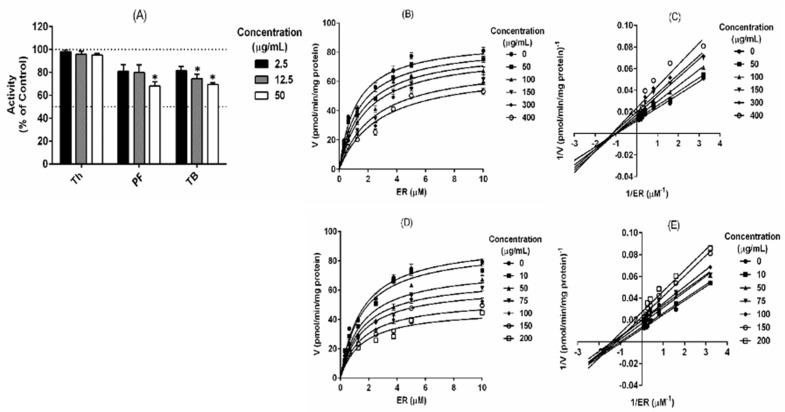
Inhibition of human recombinant CYP1A1 activity and kinetics by the tested products. (**A**) The fluorescence was recorded every 15 s during 15 min; reactions consisted in 40 µg protein, 0.32–10 μM 7-ethoxyresorufin, and 50 mM NADPH. For inhibition assays, the products were added at different concentrations to the reaction mixture. Each point in (**A**) represents the mean ± SD from three independent experiments. (**B**–**E**) Lineweaver–Burk plot analysis was done to obtain the kinetic parameters. (**B**,**C**) polyphenolic extract; (**D**,**E**) thalassiolin B. Th, *Thalassia testudinum* hydroethanolic extract; PF, polyphenolic fraction; TB, thalassiolin B. * Statistical differences (*p* < 0.05).

**Table 1 marinedrugs-18-00566-t001:** Rat CYP1A1/2 activity modulation by *T. testudinum* extract and its components.

CYP	Product	Test Product Concentrations (µg/mL)				
**rCYP1A1**		**2.5**	**5.0**	**12.5**	**25.0**	**50.0**
	**Th**	74.88 ±2.92 ^a^	63.11 ±8.22 ^a,^*	49.68 ±8.04 ^a,^**	ND	ND
	**PF**	46.61 ±3.10 ^b,^**	43.31 ±0.94 ^b,^**	31.76 ±6.64 ^b,^**	27.07 ±1.95 ^a,^**	24.00 ±2.97 ^a,^**
	**TB**	53.83 ±1.08 ^c,^*	42.97 ±4.66 ^b,^**	42.35 ±0.84 ^a,^**	29.99 ±3.03 ^a,^**	35.15 ±1.92 ^b,^**
**rCYP1A2**		**2.5**	**5.0**	**12.5**	**25.0**	**50.0**
	**Th**	100.47 ±10.66 ^a^	99.88 ±5.25 ^a^	94.65 ±14.80 ^a^	ND	ND
	**PF**	83.93 ±2.55 ^a^	71.87 ±2.85 ^b^	62.05 ±4.92 ^b,^*	55.02 ±2.84 ^a,^*	56.02 ±2.04 ^a,^*
	**TB**	81.01 ±8.06 ^a^	75.04 ±2.78 ^b^	57.08 ±4.70 ^b,^*	63.36 ±4.90 ^a,^*	57.03 ±4.07 ^a,^*

Test products were added at 2.5–50.0 µg/mL to the incubation mixture containing rat liver microsomes (80 µg) and 7-ethoxyresorufin (1 µM) for CYP1A1 or 7-methoxyresorufin (5 µM) for CYP1A2. The values represented the means ± SD of CYP activities (% respect control) from three independent experiments. Each sample was running by triplicate. The enzymatic activity in absence of test products was taken as 100%. rCYP1A1, rCYP1A2: rat CYPs; Th: *T. testudinum* extract; PF: polyphenolic fraction; TB: thalassiolin B. Different letters (^a,b,c^) represent statistical differences (*p* < 0.05) between test products; * *p* < 0.05, ** *p* < 0.01 when compared with control (100% enzyme activity).

**Table 2 marinedrugs-18-00566-t002:** Kinetics parameters for rat and human CYP1A1 inhibition.

Inhibitor	Parameter	Rat CYP1A1	Human CYP1A1
**ER**	Vmax (pmol/min/mgPr)	2396.00 ± 116.20	95.20 ± 8.14
	Km (μM)	0.42 ± 0.05	0.34 ± 0.02
**Th**	Type of inhibition	Mixed	-
	Ki (μg/mL)	54.16 ± 9.09	-
	α	8.66 ± 2.82	-
**PF**	Type of inhibition	Mixed	Mixed
	Ki (μg/mL)	5.96 ± 1.55	197.10 ± 63.40
	α	370.60 ± 56.86	7.14 ± 5.67
**TB**	Type of inhibition	Mixed	Non-competitive
	Ki (μg/mL)	3.05 ± 0.89	203.10 ± 17.29
	α	3.65 ± 0.86	-

Data represent the mean ± SD of three independent experiments. Kinetic parameters were obtained by a nonlinear regression analysis of experimental data fitted to Michaelis-Menten equation. EROD (7-ethoxyresorufin) was used as substrate, Vmax: maximum velocity, Km: Michaelis-Menten constant. Th: *T. testudinum* hydroethanolic extract; PF: polyphenolic fraction; TB: thalassiolin B. Human recombinant CYP1A1 was obtained from *E. coli*.

**Table 3 marinedrugs-18-00566-t003:** Effects of *T. testudinum* extract and its components on benzo[a]pyrene-induced mutagenicity in *Salmonella typhimurium*.

Treatments	His^+^ Revertants/Plate (% Inhibition)
S_9_-Control	23.1 ± 2.0
S_9_-Control vehicle + BP	746.2 ± 32.3
S_9_-Th (10 µg/mL) + BP	714.4 ± 8.5
S_9_-Th (100 µg/mL) + BP	678.0 ± 11.2 ** (10%)
S_9_-Th (1000 µg/mL) + BP	547.4 ± 5.9 *** (27%)
S_9_-PF (5 µg/mL) + BP	713.4 ± 8.8
S_9_-PF (50 µg/mL) + BP	665.5 ± 10.1 ** (11%)
S_9_-PF (500 µg/mL) + BP	497.3 ± 10.7 *** (34%)
S_9_-TB (400 µg/mL) + BP	512.1 ± 28.4 *** (32%)

Data represent mean ± SD of histidine revertant colonies number in TA98 *S. typhimurium* strain of two independent experiments by triplicate. Incubations were in presence of rat liver microsomal mix (S9). Th: *T. testudinum* extract; PF: polyphenolic fraction; TB: thalassiolin B, BP: benzo[a]pyrene. ** *p* < 0.01, *** *p* < 0.001, ANOVA followed by Tukey test, compared to control vehicle + BP. %inhibition: percentage of inhibition number revertants/plate in regards to “S9-Control (vehicle) + BP” group, as (1-(colonies/plates with BP + product)/(colonies/plate with just BP)) × 100%.

**Table 4 marinedrugs-18-00566-t004:** Effects of *T. testudinum* extract pre-treatment on oxidative stress biomarkers in mice after exposure to benzo(a)pyrene.

Treatment (mg/kg)	MDA (µM/mgPr)	AOPP (µM chloramines/mgPr)	GSH (µM/mgPr)
**Control (vehicle)**	3.17 ± 0.5	7.41 ± 1.3	495.1 ± 67.8
**BP**	7.31 ± 0.2 ^a^	14.67 ± 1.2 ^a^	149.7 ± 63.5 ^a^
**Th + BP**	5.55 ± 0.6 ^b^	11.64 ± 1.5 ^b^	352.8 ± 35.1^b^

Values are expressed as mean ± SD (concentration *per* mg protein). ^a^ significant difference regarding control, ^b^ significant difference regarding to BP group. Control: animals received 7 daily oral doses of distilled water and one dose of oil (BP vehicle). Th: *T. testudinum* extract, BP: animals received 250 mg/kg benzo(a)pyrene, BP + *T. testudinum:* 7 days oral pre-treatment with 100 mg/kg *T. testudinum* extract before receiving BP dose. ANOVA-Dunnett post hoc-test, *p* < 0.05. MDA: malondialdehyde, AOPP: advanced oxidation protein products, GSH: reduced glutathione.

**Table 5 marinedrugs-18-00566-t005:** Effects of *T. testudinum* extract on benzo(a)pyrene-induced micronucleus in mice bone marrow.

Treatment (mg/kg)	PCE/NCE	MN/PCE
Control (vehicle)	1.8 ± 0.26 ^b^	4.0 ± 0.7 ^b^
BP	3.4 ± 0.78 ^a^	17.0 ± 1.7 ^a^
Th + BP	2.0 ± 0.21 ^b^	7.0 ± 1.0 ^a,b^

Values are expressed as mean ± SD. MN: micronucleus, PCE: polychromatic erythrocytes, NCE: normochromatic erythrocytes (2000 cells/animal), Control: Animals received 7 daily oral doses of distilled water and one dose of oil (BP vehicle). Th: *T. testudinum* extract, BP: animals received 250 mg/kg of benzo(a)pyrene, BP + *T. testudinum*: 7 days’ oral pre-treatment with 100 mg/kg *T. testudinum* extract before receiving BP dose, ^a^ significant difference regarding control, ^b^ significant difference regarding to BP group. ANOVA–Dunnett post hoc-test, *p* < 0.05.

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
