# Peer review of "Interaction of Thalassia testudinum Metabolites with Cytochrome P450 Enzymes and Its Effects on Benzo(a)pyrene-Induced Mutagenicity"

_marinedrugs, 2020, doi:10.3390/md18110566_

Round 1

Reviewer 1 Report

The manuscript "Interaction of Thalassia testudinum metabolites with 2 Cytochrome P450 enzymes and its effects on 3 benzo(a)pyrene-induced mutagenicity " by Livan Delgado-Roche et al. is devoted to the study of Thalassia testudinum hydroethanolic extract, its polyphenolic fraction, and thalassiolin B impact on the activity of phase I metabolizing enzymes and their antimutagenic effects. Presented study claims that the metabolites of T. testudinum may represent a potential source of chemopreventive compounds for adjuvant therapy of cancer.

In general, the manuscript presents interesting data, is well organized and documented, and is written in acceptable form.

Besides quality and perception improving of the manuscript I would suggest to pay attention to the following notes and minor remarks:

  • To my point of view, it would be worth to indicate the chemical structure of the tested 1-chrysoeriol 7-β-D-glucopyranosyl-2″-sulphate in the text;
  • 9, lines 315-316: the spectroscopic data confirming the structure of 1-chrysoeriol 7-β-D-glucopyranosyl-2″-sulphate should be present in the experimental part;
  • The Conclusion section should be extended and include more detailed results derived from the presented study.

My decision is minor revision.

Author Response

Dear Reviewer,

We appreciate your time to review our manuscript. Please, find below our response to comments.

The manuscript "Interaction of Thalassia testudinum metabolites with 2 Cytochrome P450 enzymes and its effects on 3 benzo(a)pyrene-induced mutagenicity " by Livan Delgado-Roche et al. is devoted to the study of Thalassia testudinum hydroethanolic extract, its polyphenolic fraction, and thalassiolin B impact on the activity of phase I metabolizing enzymes and their antimutagenic effects. Presented study claims that the metabolites of T. testudinum may represent a potential source of chemopreventive compounds for adjuvant therapy of cancer.

In general, the manuscript presents interesting data, is well organized and documented, and is written in acceptable form. Besides quality and perception improving of the manuscript I would suggest paying attention to the following notes and minor remarks:

To my point of view, it would be worth to indicate the chemical structure of the tested 1-chrysoeriol 7 β-D-glucopyranosyl-2″-sulphate in the text; lines 315-316: the spectroscopic data confirming the structure of 1-chrysoeriol 7-β-D-glucopyranosyl-2″-sulphate should be present in the experimental part.

The chemical structure of 1-chrysoeriol 7 β-D-glucopyranosyl-2″-sulphate was previously characterized by our group. The spectroscopic data was included as suggested by the reviewer, please see Figure 1.

Reference:

- Garateix A, Salceda E, Menéndez R, Regalado EL, López O, García T, et al. (2011). Antinociception produced by Thalassia testudinum extract BM-21 is mediated by the inhibition of acid sensing ionic channels by the phenolic compound thalassiolin B. Mol. Pain. 7, 1–15. doi.org10.1186/1744-8069-7-10

The Conclusion section should be extended and include more detailed results derived from the presented study.

My decision is minor revision.

Thank you very much for your valuable comments.

Dr. Rodeiro

Dr. Espinosa-Aguirre

Reviewer 2 Report

In their manuscript entitled “Interaction of Thalassia testudinum metabolites with 3 Cytochrome P450 enzymes and its effects on 4 benzo(a)pyrene-induced mutagenicity”, Delgado-Roche and colleagues report the investigation of the effects of a hydroethanolic extract, a polyphenolic fraction, and of a thalassiolin B preparation from the marine plant Thalassia testudinum on the activity of phase I metabolizing enzymes CYP1A1/2 and CYP2B1/2 activity. The authors also have investigated their antimutagenic effect against benzo[a]pyrene in a mice model. The work is well organized and written in an acceptable way. The use of too many abbreviations difficult at some point the reading of the manuscript and there are some abbreviations used without an explanation, which can easily be fixed by the authors. However, there are some issues that should be considered. 1) The authors state throughout the manuscript that rat and human CYP1A1/2 CYP2B1/2 were used. However, after reading the materials and methods section it becomes clear that human enzymes were in fact recombinant enzymes produced in E.coli. The authors should clearly state throughout the work and whenever adequate that the human enzymes were produced in E. coli as recombinant proteins. 2) based on the results presented in Table 1, the authors conclude that CYP1A1 was more sensitive than CYP1A2 to the hydroethanolic extract (Th), the polyphenolic fraction (PF), and the thalassiolin B preparation (TB). A control should be presented in Table 1, with the activities determined in the absence of extracts. In addition, since the activities are presented as absolute values, it is hard for the reader to easily understand this conclusion. I suggest that the conclusion is accompanied with a brief description on how this conclusion was reached; 3) Figure 2A is hard to understand, as there is a set of 3 bars at the center of the figure for which apparently thee is a label missing. PF? For Th, a slight inhibition is shown, but the authors state that no inhibition was detected. This should be corrected; 4) In line 153 the authors state that Th did not inhibit the CYP1A1 enzyme, and in lines 157-158 it is stated that “These results showed a potential inhibitory effect of phase I carcinogen-metabolizing 158 enzymes CYP1A1 by T. testudinum extract and its components”. The sentences should are contradictory, and should be rephrased. Minor issues: The names of organisms should be italicized throughout the text, tables, and figures legends; lines 189-190, the abbreviation AOPP should be explained as it appears here for the first time without any explanation; line 189, the same for MDA; lines 220-221, the sentence “is rich in flavonoids” is vague, can the authors give a number, or a characterization of the flavonoid content?; line 221-223, a reference should be provided; line 308, please detail how the leaves were dried; line 330, “d ultracentrifugation steps” ? please detail; line 384, AOPP should be expressed as micromolar per mg protein, the same applies for the other parameters, as the amount of protein in the reactions mixtures is not necessarily exactly the same; line 391, correct as follows: “was determined…”.

Author Response

Dear Reviewer,

We appreciate your time to review our manuscript. Please, find below our responses point by point to your comments.

In their manuscript entitled “Interaction of Thalassia testudinum metabolites with 3 Cytochrome P450 enzymes and its effects on 4 benzo(a)pyrene-induced mutagenicity”, Delgado-Roche and colleagues report the investigation of the effects of a hydroethanolic extract, a polyphenolic fraction, and of a thalassiolin B preparation from the marine plant Thalassia testudinum on the activity of phase I metabolizing enzymes CYP1A1/2 and CYP2B1/2 activity. The authors also have investigated their antimutagenic effect against benzo[a]pyrene in a mice model.

- The work is well organized and written in an acceptable way.

- The use of too many abbreviations difficult at some point the reading of the manuscript and there are some abbreviations used without an explanation, which can easily be fixed by the authors.

Some abbreviations were abrogated in order to allow an easier text comprehension. As recommended, the rest of the abbreviations were defined or explained once appear by the first time in the text.

- However, there are some issues that should be considered.

1) The authors state throughout the manuscript that rat and human CYP1A1/2 CYP2B1/2 were used. However, after reading the materials and methods section it becomes clear that human enzymes were in fact recombinant enzymes produced in E.coli. The authors should clearly state throughout the work and whenever adequate that the human enzymes were produced in E. coli as recombinant proteins.

The recombinant origin of human CYP1A1 was clarified in the text of the revised manuscript when considered necessary as recommended by the reviewer.

2) based on the results presented in Table 1, the authors conclude that CYP1A1 was more sensitive than CYP1A2 to the hydroethanolic extract (Th), the polyphenolic fraction (PF), and the thalassiolin B preparation (TB). A control should be presented in Table 1, with the activities determined in the absence of extracts. In addition, since the activities are presented as absolute values, it is hard for the reader to easily understand this conclusion. I suggest that the conclusion is accompanied with a brief description on how this conclusion was reached.

As we described in the section of Methods, the percentage values of CYP activities were calculated vs a control reaction mixture without any extract or potential inhibitor (100% of CYP activity). Therefore, we consider unnecessary include the 100% of CYP activity in Table 1.  

On the other hand, the statistical analysis showed that 30% of inhibition or less was considered a significant inhibitory effect. Therefore, we concluded that CYP1A1 was more sensitive than CYP1A2 to Th, PF and TB based on the fact that from the lowest concentrations (2.5 µg/mL), the extracts significantly inhibited CYP1A1 activity, not for CYP1A2, where greater concentrations were needed to achieve a significant inhibition of enzymatic activities (25 µg/mL as mean).

3) Figure 2A is hard to understand, as there is a set of 3 bars at the center of the figure for which apparently thee is a label missing. PF? For Th, a slight inhibition is shown, but the authors state that no inhibition was detected. This should be corrected.

The figure 2A was corrected, the label of PF was included. In addition, the description of the results was corrected as reviewer recommended:

“…while Th only exhibited a slight inhibitory effect on enzyme activity at the highest concentration…”

4) In line 153 the authors state that Th did not inhibit the CYP1A1 enzyme, and in lines 157-158 it is stated that “These results showed a potential inhibitory effect of phase I carcinogen-metabolizing 158 enzymes CYP1A1 by T. testudinum extract and its components”. The sentences should are contradictory, and should be rephrased.

The sentences were rephrased as: “These results showed a potential inhibitory effect of phase I carcinogen-metabolizing enzymes CYP1A1 by T. testudinum metabolites.”

Minor issues:

The names of organisms should be italicized throughout the text, tables, and figures legends.

The names of organisms were italicized throughout the text.

Lines 189-190, the abbreviation AOPP should be explained as it appears here for the first time without any explanation. Line 189, the same for MDA.

These abbreviations were defined in the text of revised manuscript as: “…the serum levels of malondialdehyde (MDA), advanced oxidation protein products (AOPP) and reduced glutathione (GSH).”

Lines 220-221, the sentence “is rich in flavonoids” is vague, can the authors give a number, or a characterization of the flavonoid content?

Our group previously determined the percentages of flavonoids and other polyphenols from the hydroethanolic extract of T. testudinum. The following text was included in the revised manuscript: “…T. testudinum marine plant is rich in flavonoids and other polyphenols (29.5% ± 1.2% total polyphenols, proanthocyanidins 21.0% ± 2.3%, total flavonoids 4.6% ± 0.2%, expressed as g per 100 g of the dry extract, % w/w) suggesting…”

Reference:

- Regalado EL, Menéndez R, Valdés O, Morales RA, Laguna A, Thomas OP, et al. (2012). Phytochemical analysis and antioxidant capacity of BM-21, a bioactive extract rich in polyphenolic metabolites from the sea grass Thalassia testudinum. Nat. Prod. Commun. 7, 47–50.

- Rodeiro I, Hernández I, Herrera JA, Riera M, Donato MT, Tolosa L, González K, et al. (2018). Assessment of the cytotoxic potential of an aqueous-ethanolic extract from Thalassia testudinum angiosperm marine grown in the Caribbean Sea. J. Pharm. Pharmacol. 70, 1553–60. doi.org/10.1111/jphp.13001

Line 221-223, a reference should be provided.

References 45-47 sustained the line 221-223.

Line 308, please detail how the leaves were dried

The treatment of T. testudinum leaves was added in the revised manuscript: “The leaves were washed with water to eliminate sediments and the excess of salt, then, the plant material was dried at room temperature.”

Line 330, “d ultracentrifugation steps” ? please detail;

The centrifugation steps were clarified as follow: “S9 fraction was obtained as previously described [57]. To obtain the microsomal fraction, S9 was split into 1 mL aliquots and centrifuged at 100 000 g and 4ºC per 60 min. The pellet was resuspended in 0.1 M phosphate buffer (pH 7.4) plus 0.25 M sucrose and it was centrifuged again at 100 000 g at 4ºC per 60 min…”

Line 384, AOPP should be expressed as micromolar per mg protein, the same applies for the other parameters, as the amount of protein in the reactions mixtures is not necessarily exactly the same;

The observation was considered and corrected in the text. The concentration of oxidative stress biomarkers was expressed per mg of protein.

Line 391, correct as follows: “was determined…”.

The phrase was corrected as recommended.

On behalf of the authors, thank you again for your valuable comments.

Kind regards.

Dr. Rodeiro

Dr. Espinosa-Aguirre

Reviewer 3 Report

Dear Editor,

Please, find bellow my comments and suggestions regarding the manuscript entitled:  Interaction of Thalassia testudinum metabolites with 2 Cytochrome P450 enzymes and its effects on 3 benzo(a)pyrene-induced mutagenicity, by Idania Rodeiro and Javier J. Espinosa-Aguirre, et al.

From the formal point of view, the manuscript is well written and structured, the English language is correct. The experimental part is described in detail and the bibliographic references are adequate and with the correct format. Some typing errors are present that should be corrected by authors, i.e.: line 320 in page 9, among others.

From the scientific point of view, the study is well developed and approached with the adequate methodologies, consistent with the target of the study. However, a more ambitious study could be developed in the future evaluating other targets and other action mechanisms.

It is mentioned in the text that some major flavonoid glycosides have been identified in the ethanol extract of Thalassia testudinum, but the study has been performed by using the complex mixture of the extract. It would be worthwhile to evaluate the activity of the isolated active principles in order to determine possible lead compounds for future pharmacological applications.

Regarding the main objective of this study, i. e., to propose the use of this extract as antioxidant to treat the adverse effects (carcinogenic effects) produced by polycyclic benzene hydrocarbons, generated by the combustion of fossil fuels (environmental pollution of big cities); it would be more interesting to focus the target of this study in the potential preventive or prophylactic treatment of smokers against the development of cancer in the respiratory tract due to polycyclic benzenes hydrocarbons present in the tar generated during the combustion of tobacco. This is of special interest in Cuba where consumption of tobacco has a large tradition.

On the other hand, apart from the study of the influence of this extract on CYP1A and CYP2B, it would be worthwhile to study other action mechanisms of these compounds in order to develop multitarget active principles.

Finally, a study on the toxicity of these compounds (extract) should be evaluated in the future, in order to evaluate whether the benefits are higher than the adverse effects.

Overall, my opinion on the manuscript is positive and my recommendation is to publish it after a minor revision, correcting the typing errors and introducing some comments on the question before mentioned.

Author Response

Dear Reviewer,

We appreciate your time to review our manuscript. Please, find below our responses (point by point) to your comments.

From the formal point of view, the manuscript is well written and structured, the English language is correct. The experimental part is described in detail and the bibliographic references are adequate and with the correct format. Some typing errors are present that should be corrected by authors, i.e.: line 320 in page 9, among others.

The English grammar and style was revised as indicated. 

From the scientific point of view, the study is well developed and approached with the adequate methodologies, consistent with the target of the study. However, a more ambitious study could be developed in the future evaluating other targets and other action mechanisms.

The authors agree with the reviewer comments. In future studies, we´ll assess the anticarcinogenic potential effects of T. testudinum extract in transgenic animal models, allowing us to determine potential mechanisms of action.

It is mentioned in the text that some major flavonoid glycosides have been identified in the ethanol extract of Thalassia testudinum, but the study has been performed by using the complex mixture of the extract. It would be worthwhile to evaluate the activity of the isolated active principles in order to determine possible lead compounds for future pharmacological applications.

As we mentioned in the manuscript, the major component of the whole hydroethanolic extract is thalassiolin B, thus we have evaluated its pharmacological activity. However, in future studies, we´ll taking into consideration the reviewer´s comments to assess potential pharmacological activities of other active metabolites. In fact, we have fractionated the whole extract of T. testudinum into a polar and non-polar fractions to test pharmacological activities (e.g. the polyphenolic fraction obtained from T. testudinum was tested in the present work).

Regarding the main objective of this study, i. e., to propose the use of this extract as antioxidant to treat the adverse effects (carcinogenic effects) produced by polycyclic benzene hydrocarbons, generated by the combustion of fossil fuels (environmental pollution of big cities); it would be more interesting to focus the target of this study in the potential preventive or prophylactic treatment of smokers against the development of cancer in the respiratory tract due to polycyclic benzenes hydrocarbons present in the tar generated during the combustion of tobacco. This is of special interest in Cuba where consumption of tobacco has a large tradition.

The authors agree with reviewer comments. Therefore, in the revised manuscript we included the tobacco consumption as an important source of PAHs, in particular BP.

In accordance with WHO estimates, lung cancer causes about 6% of premature deaths related to outdoor air pollution, as well as tobacco consumption [1]. Polycyclic aromatic hydrocarbons (PAHs) are pollutants widely distributed in the environment as a result of organic matter incomplete combustion. Also, PAHs are present in commercial products consumed by humans (e.g. tobacco) [2].

On the other hand, apart from the study of the influence of this extract on CYP1A and CYP2B, it would be worthwhile to study other action mechanisms of these compounds in order to develop multitarget active principles.

The authors agree with reviewer comments. In future studies, we´ll explore other potential pharmacological targets of T. testudinum metabolites.

Finally, a study on the toxicity of these compounds (extract) should be evaluated in the future, in order to evaluate whether the benefits are higher than the adverse effects.

Studies from the literature have evaluated the acute and subchronic toxicity of Thalassia testudinum extract by oral route in mice and rats. Acute toxicity test has conducted by oral route at doses between 50 and 2000 mg/kg in both sexes’ mice. Subchronic studies in rats included three doses (100, 500 and 2000 mg/kg/day) by oral route for 90 days. Acute toxicity test revealed any signs of toxicity. The approximate lethal dose of the extract was higher than 2000 mg/kg. Dose dependent increase of urea and decrease of creatinine were observed in treated male animals in the subchronic study. These clinical signs did not represent any functional impairment for test animals and were reversible.

Reference:

Lagarto A, et al. (2020), “Data for: Safety evaluation of seagrass Thalassia testudinum extract in acute and subchronic oral toxicity studies.”, Mendeley Data, V1, doi: 10.17632/zkm7p6fvgr.1

Overall, my opinion on the manuscript is positive and my recommendation is to publish it after a minor revision, correcting the typing errors and introducing some comments on the question before mentioned.

Again, thank you very much for your valuable comments.

Kind regards,

Dr. Rodeiro

Dr. Espinosa-Aguirre

Round 2

Reviewer 2 Report

The major issues raised to the previous version of the manuscript were solved by the authors. The revised manuscript can now be accepted for publication.